# Properties of Superelastic Nickel–Titanium Wires after Clinical Use

**DOI:** 10.3390/ma16165604

**Published:** 2023-08-13

**Authors:** Inés Alcaraz, Javier Moyano, Ariadna Pàmies, Guillem Ruiz, Montserrat Artés, Javier Gil, Andreu Puigdollers

**Affiliations:** 1Department of Orthodontics, School of Dentistry, Universitat Internacional de Catalunya, Sant Cugat del Vallés, 08195 Barcelona, Spain; inesalcaraz@uic.es (I.A.); jmoyano@uic.es (J.M.); apamies@uic.es (A.P.); guillemruiz@uic.es (G.R.); martes@uic.es (M.A.); 2Bioengineering Institute of Technology, Facultad de Medicina y Ciencias de la Salud, c/Josep Trueta s7N, Sant Cugat del Vallés, 08195 Barcelona, Spain; xavier.gil@uic.es

**Keywords:** NiTi, superelasticity, sterilization, mechanical properties

## Abstract

The aim of the present study was to describe and determine changes in the superelastic properties of NiTi archwires after clinical use and sterilization. Ten archwires from five different manufacturers (GAC, 3M, ODS, GC, FOR) were cut into two segments and evaluated using a three-point bending test in accordance with ISO 14841:2006. The center of each segment was deflected to 3.1 mm and then unloaded to 0 N to obtain a load–deflection curve. Deflection at the end of the plateau and forces at 3, 2, 1 and 0.5 mm on the unloading curve were recorded. Plateau slopes were calculated at 2, 1 and 0.5 mm of deflection. Data obtained were statistically analyzed to determine differences (*p* < 0.001). Results showed that the degree of superelasticity and exerted forces differed significantly among brand groups. After three months of clinical use, FOR released a greater force for a longer activation period. GC, EURO and FOR archwires seemed to lose their mechanical properties. GC wires released more force than other brand wires after clinical use. Regarding superelasticity after sterilization, GAC, 3M and FOR wires recovered their properties, while EURO archwires lost more.

## 1. Introduction

Tooth movements and changes in orthodontic appliances result from force systems and tissue responses to them. An ideal force produces tooth movement without damaging teeth or tissues. Factors including tooth size and type of movement need to be considered when applying force during orthodontic treatment. However, it is difficult to determine an ideal force [1,2]. Hence, a sound knowledge of the mechanical behavior of orthodontic archwires is required to select the most suitable size and material to achieve optimal and predictable treatment results [3]

Orthodontic treatment begins with leveling and aligning the teeth with nickel–titanium (NiTi) archwires, which deliver light and continuous forces for efficient tooth movement [4,5,6]. NiTi archwires, introduced by Andreasen and Hillman in the 1970s [4,5], deliver an optimal constant force over an extended range of deflection in order to enable a smooth transformation into and from a martensitic phase [1,7,8]. Their behavior is characterized by a more or less constant curve of deflection load and by a more or less constant release of the force, depending on the manufacturers. This phase transformation may occur through variations in temperature and stress changes in the oral cavity [1,9].

The shape memory of Superelastic NiTi archwires is incorporated in the production phases by establishing a shape at approximately 482 °C [1]. It has been suggested that nickel–titanium alloys become ductile and may be plastically deformed at temperatures below the transition temperature range (TTR). Another property of NiTi archwires is their superelasticity or pseudoelasticity, which is produced by reversible phase transformation from the body-centered cubic structure (austenitic) to a monoclinic structure (martensitic). This phase transformation is the effect of the stress application during activation and deactivation [6]. During transformation, the stress remains stable even as it increases, preventing undesirable side effects such as hyalinization, pain, and root resorption [10,11]. As shown in the load/deflection diagram (Figure 1), superelasticity is characterized by the plateau, indicating that the force exerted is relatively constant in the range of tooth movement.

The first NiTi archwire wires in orthodontic treatments are usually used for at least two or three months, during which they undergo mechanical stress and chemical exposure in the patient’s mouth. Furthermore, studies indicate that 52% of clinicians recycle these wires [12]. During sterilization, recycled wires endure further stress [3,12,13]. Hence, repeated exposure and sterilization may induce changes in the mechanical properties and surface conditions of NiTi archwires that may not be clinically adequate for tooth movement [12].

It should be noted that the properties of NiTi orthodontic wires may depend on various aspects; chief among them are consistent chemical composition [14,15], grain size [14,16], mechanical and thermal cycling [17], residual stresses [18], and whether the surface treatment of wires improves friction coefficients. In addition, possible heat treatments of wires for flexibility, loops, or soldering to obtain different forces in the molar or canine areas produce substantial changes in the transformation temperatures and, therefore, in the superelastic curves [18,19]. Numerous studies have demonstrated that changes in temperature cause precipitation of precipitates rich in nickel or titanium, depending on the chemical composition of the wire, lead to a loss of the superelastic properties, rendering the wires useless for orthodontic therapy [20,21,22]. There appears to be limited evidence on the intraoral aging sequence and associated changes in NiTi archwire properties. Some studies have reported that heat sterilization has detrimental effects on the elastic and tensile properties of these wires [23], but more studies tend to focus on corrosion resistance and other effects on mechanotherapy. However, few reports in the last years on changes in superelastic properties have been found [24,25].

The aim of the present study was to determine changes in the superelastic properties of NiTi archwires after clinical use and sterilization and identify possible differences among the different archwire brands.

## 2. Materials and Methods

The sample comprised preformed NiTi wires from five manufacturers (GC Orthodontics Europe GmbH, Breckerfeld, Germany; Unitek, Monrovia, CA, USA; Ods, Kisdorf, Germany; Forestadent, Pforzheim, Germany; and GAC, Grenoble, France) commercialized as superelastic with a 0.016-inch round section, as shown in Table 1. The archwires were randomly distributed among orthodontic patients of the clinic at Universitat Internacional de Catalunya (UIC Barcelona, Barcelona, Spain). The study was approved by the Clinical Research Ethics Committee of the Universitat Internacional de Catalunya (ORT-ELM-2019-01). The sample was divided into three groups of archwires per brand at the following time points: 10 archwires before clinical use (T0); 10 after three months in the mouth (T1); and 10 after three months in the mouth and sterilization (T2). Each archwire was cut into two samples to test 20 segments in each group. Each of the five brands was tested 60 times, totaling 300 specimens.

Accepting an alpha risk of 0.05 and a beta risk of 0.2 in a two-sided test, twenty specimens per brand were necessary to recognize a difference greater than or equal to 0.05 units as statistically significant. The standard deviation was assumed to be 0.309, as in previous studies [1,7].

The three-point elastic bending test was used to assess the mechanical properties using a 10 mm beam length [25]. The distance between the penetrator point (Figure 2 (1^a^)) and the two supporting points of each archwire was identical (Figure 2 (2^a^)). The radius of these two points measured 0.10 ± 0.05 mm, in accordance with the ISO 14841:2006 (Figure 2) [25,26].

Each wire segment was tested once using the TestXpert III (Z005 Test Control II, Universal Testing Machine, Zwick Roell, Kennesaw, GA, USA). Sterilized samples were carried out with Standard Autoclave Sterilization (Matachana Series S1000 Sterilizers, Matachana Group S.A, Selmsdorf, Germany) at 134 °C and a pressure of 15.0 kPa for 20 min. The middle portion of the wire was deflected at a crosshead speed of 7.5 mm/min under the pressure of the penetrator point. The middle portion of each segment was loaded to 3.1 mm of deflection, similar to what is usually produced in clinical situations when teeth are levelled and aligned. The wire segments were unloaded at the same crosshead speed until the released force reached zero. Subsequently, the unloading curve and the superelasticity of each wire segment were evaluated using the nine following parameters:Force level delivered in Newtons (N) with deflection at 3.0 mm, 2.0 mm, 1.0 mm, 0.5 mm (Fdef-3 mm, Fdef-2 mm, Fdef-1 mm, Fdef-0.5 mm, respectively).Deflection at the end of plateau in mm (Sp).Minimum force level at the end of superelastic plateau in N (Fp).Plateau slopes: between 0.5 mm and Sp (Slope-0.5 mm), between 1 mm and Sp (Slope-1 mm), and between 2 mm and Sp (Slope-2 mm) of deflection expressed in N/mm.

The plateau slope value was obtained using the following equations. The equation $A.\;\mathrm{Slope}-0.5\mathrm{mm}\;\left(N/g\right)=\frac{\left(\mathrm{Fdef}-0.5\mathrm{mm}\right)-\mathrm{Fp}}{\left(0.5-\mathrm{Sp}\right)}$ shows the slope at 0.5 mm of deactivation; the equation $B.\;\mathrm{Slope}-1\mathrm{mm}\;\left(N/g\right)=\frac{\left(\mathrm{Fdef}-1\mathrm{mm}\right)-\mathrm{Fp}}{\left(1-\mathrm{Sp}\right)}$ indicates the slope at 1 mm of deactivation; finally, the equation $C.\;\mathrm{Slope}-2\mathrm{mm}\;\left(N/g\right)=\frac{\left(\mathrm{Fdef}-2\mathrm{mm}\right)-\mathrm{Fp}}{\left(2-\mathrm{Sp}\right)}$ shows the slope at 1 mm of deactivation measuring the degree of plateau flatness; thus, the closer the slope is to zero, the more constant the force. Figure 1 shows the loading and unloading curve of the archwire segments and all the evaluated parameters.

The results were analyzed with R software version 4.2.1 (1989, 1991 Free Software Foundation, Inc. Temple Place, Boston, MA, USA). The variables of interest were first described by mean and standard deviation. Normality was tested with the Shapiro–Wilk test. The groups were compared with ANOVA. All tests were considered significant for a *p*-value of less than 0.05. Statistically significant differences were set at a *p*-value < 0.001. Further, clustering analysis was performed on each group in order to know its behavior. That means all the performances in cluster A have similar behavior, the same as those in clusters B, C, and D.

The microstructures of the NiTi studied were observed using an SEM (JEOL 1200 EXII Microscopy Tokyo, Japan) equipped with a link LZ5 EDS (Jeol, Tokyo, Japan), which was used for determining the chemical composition. Original samples were etched with HF 35% *v*/*v* for 15 s in order to reveal the microstructure. The same acid etching was realized for the samples with stabilized martensite. An equiatomic NiTi was heat-treated at 550 °C for 60 min, in this case, the samples were not etched.

## 3. Results

Graphs were made showing the behavior of each wire at the three time points (Figure 3, Figure 4 and Figure 5). Note the comparable shape and an apparent superelastic plateau, albeit an apparent discrepancy in the loading and unloading curve.

Table 2 and Table 3 show the Plateau Slopes at 0.5 mm, 1 mm, and 2 mm of deactivation and the mean delivered force (Fdef) at 3, 2, 1, and 0.5 mm deflection, respectively. The variations in properties of wire segments before clinical use (T0) and after three months (T1) were compared; changes among T1, T2, and T0 were also analyzed. Additionally, cluster distribution is shown in order to notice the different conditions of each group.

## 4. Discussion

The present study has a twofold aim: to analyze the superelastic properties of NiTi wires at three time points and to determine the changes in the mechanical properties after recycling the wires used in patients and non-in vitro simulation processes (thermocycling, cyclic loading). There appear to be limited clinical studies and a notorious inability of in vitro research to simulate in vivo conditions [27,28,29] since the multiplicity of factors present in the oral cavity cannot be simulated [27,30,31,32]. Due to its reproducibility, the three-point bending test is the standard method for testing and comparing flexural properties [3,5,33,34]. However, studies show wide variability in this method, indicating no established consensus about how testing should be undertaken [3,5,9,35]. Currently, this bending test is regulated by two standards: the European National Standard and the American National Standard (ANSI/ADA), which are identical to ISO 15841:2006 [23,26,36]. Thus, it is recommended to follow the same standard procedure with equally adjusted parameters to accurately compare archwire properties. Archwire manufacturers are required to indicate the properties of their product according to the three-point bending test, as set out in the European National Standard [26,37]. The testing procedure used in the present study was conducted according to ISO 15841:2006 [26]. However, most previous studies have not followed these criteria [6,13,33,38,39,40,41]. As a result, only two articles have been found in which testing was performed according to ISO 15841:2006 [15,35].

Nine parameters were studied for the three groups of NiTi archwires supplied by five brands. The plateau slopes calculated to determine the superelastic properties of each wire revealed great variability (Figure 3, Figure 4 and Figure 5) [5,15]. The statistical differences among manufacturers for the means of each variable were measured at T0 (before clinical use). The GC archwires exerted the highest forces at all deflection points, although the FOR and 3M archwires behaved similarly. Thus, at T1 (after three months of use in the mouth), the FOR group exerted the highest forces at all deflection points, except at 2 mm, where GC and 3M wires performed similarly to the FOR. At T2 (after sterilizing the used archwires), the GC group exerted the highest force at three deflection points, except at 3 mm, where the 3M archwires behaved identically to the GC. Despite the differences observed at T0, all the archwires exerted almost the same force after three months of use (Table 2). Some authors point out that force increases as crowding is resolved but then recovers baseline properties once teeth are aligned [23,33].

Plateau slope values show the degree of superelasticity. If these values remain stable over time, superelastic properties remain stable. A constant force over a wide range allows the orthodontist to use the archwire even on teeth far from their correct position in the arches. The present study found significant differences in plateau slope values across the studied groups (Table 3). Regarding lower deformations (0.5–1 mm), GAC and 3M archwires did not remain stable after three months, lowering values and becoming more superelastic, after which sterilization plateau values remained constant. As for lower deformations, EURO, GC, and FOR at the plateau increase, making the archwires gradually lose their superelasticity after three months in the oral cavity. After sterilization, the plateau slope values also increased, compromising superelasticity. In contrast, the FOR wires showed no decrease in plateau values, thus recovering superelastic properties [9].

Despite these differences among the archwire brands at the different time points, we found that all the groups released almost the same force as before clinical use at all deflection points (Table 3). This may be due to the fact that as crowding resolves, as observed in other studies [7,9], the force initially increases but subsequently recovers its baseline properties. These results suggest that the wire is subjected to continuous deformations due to crowding. Once crowding is resolved, the archwire stops receiving continuous activation forces as it recovers its initial properties [9].

The superelastic properties observed after months of treatment can be attributed to the presence of martensite that is stabilized by mechanical stress. When the clinician inserts an orthodontic wire, the wire is in the initial phase known as the austenitic phase. However, due to the tension caused by the misalignment of the teeth, the wire exerts a corrective force to realign the dental position. This force exerted by the wire on the teeth results from a martensitic transformation induced by the stress in the region of maximum tension [42].

The martensitic phase induced by the tension is not stable at the oral cavity temperature of 37 °C. Consequently, under mechanical tensions, it tends to revert back to the original phase and restore the initial arch shape, which is considered optimal for each patient.

These transformations are known as thermoelastic martensitic transformations, as they do not result in plastic deformations like those observed in steel. Instead, they allow for wires with superelastic behavior that can withstand deformations exceeding 20%. However, when the wires remain in service for an extended period or experience high loads, dislocations occur that anchor the martensite plates. These plates, once stabilized, no longer return to their initial phase at 37 °C. To unanchor these plates, significantly higher temperatures are required, typically ranging from 400–500 °C, as indicated by several authors. The stabilization of the martensite inhibits superelasticity and diminishes the effectiveness of the orthodontic wire, a phenomenon referred to as NiTi wire amnesia.

Figure 6A shows self-accommodating martensite plates resulting from wire cooling and at higher magnification, while Figure 6B shows martensitic plates with various orientations. As the temperature does not have a vectorial orientation, martensite plates form in the most favorable orientations. This is called martensitic self-accommodation. However, Figure 6C illustrates martensite plates induced by mechanical stress, revealing that these stress-induced plates align with the direction of mechanical stress. Unlike temperature, which is not a vectorial property, stress is a vectorial parameter, thereby influencing the direction of the plates. Figure 6C shows the martensitic plates that have stabilized within a wire, which was initially austenitic at room temperature. However, after a stress period of 3 months, it is evident that a part of the martensitic plates did not transform back to the austenitic phase. The stabilized martensite partially or totally loses the superelasticity of the orthodontic wire. In order to retransform stabilization to austenite and recover the superelasticity of the austenite, the wire must be subjected to a heat treatment of approximately 400 °C for 20 min. Care must be taken not to increase the temperature and treatment times as this can lead to the appearance of titanium-rich precipitates in the austenitic matrix which cause the superelasticity of the wire to be lost (Figure 6D).

Before starting orthodontic treatment, it is crucial to consider the changes in mechanical properties of each brand, in particular superelasticity and delivered forces, which affect crowding resolution and periodontal conditions. It is also relevant that some archwires recover their initial properties after sterilization, allowing them to be reused.

## 5. Conclusions

Significant differences in exerted forces, both during three months of activation (T1) and after sterilization (T2), were found among the brands.GC, EURO, and FOR appeared to lose their superelastic properties during three months of clinical use (T1).As for lower deformations (<2 mm), GAC, 3M, and FOR wires recovered their properties after sterilization (T2), while EURO archwires appeared to lose their superelasticity.Superelastic properties and released forces were found to differ significantly among all groups studied (T0, T1, and T2).

## Figures and Tables

**Figure 1 materials-16-05604-f001:**
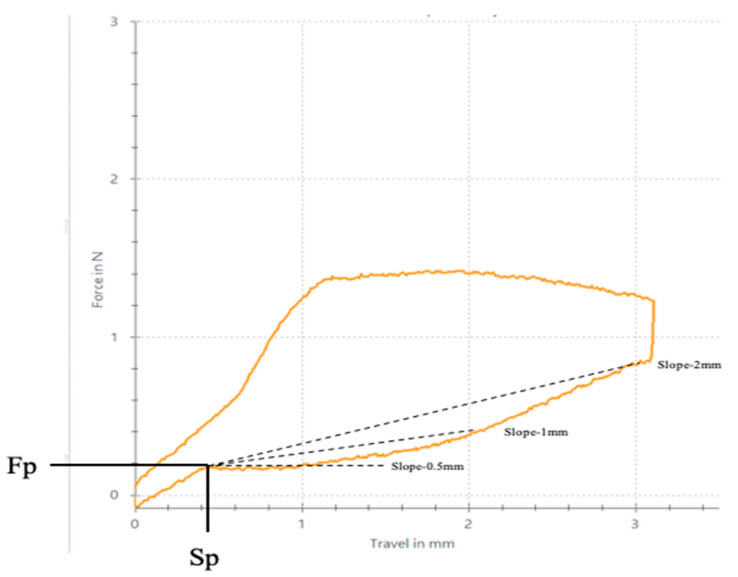
Loading and unloading curve of NiTi archwire and all evaluated parameters of the study.

**Figure 2 materials-16-05604-f002:**
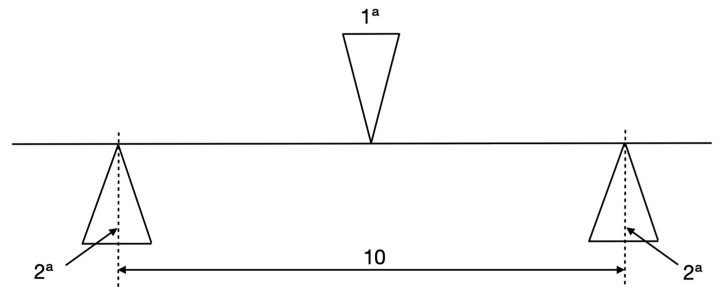
Sketch of the procedure of the three-point bending test according to ISO 14841:2006, with a radius of support and penetrator points 0.10 ± 0.05 mm. 1^a^. Loading Force/ Penetrator point. 2^a^ Two parallel supporting points.

**Figure 3 materials-16-05604-f003:**
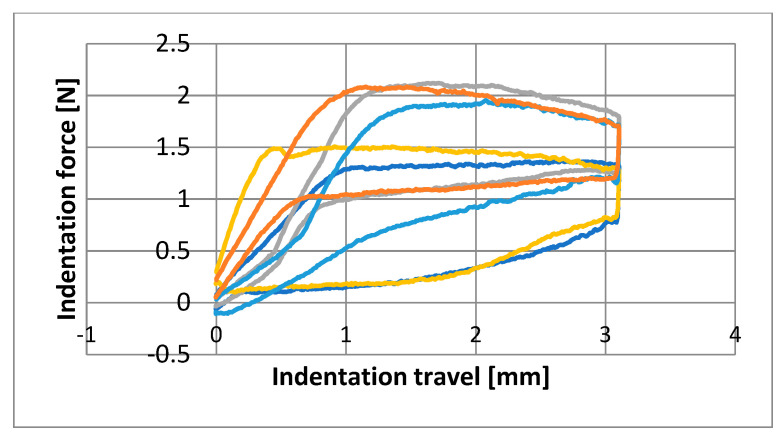
Diagram of the first specimen from each manufacturer obtained from as received archwires before clinical use (T0). Dark blue: GC/Blue: 3M/Yellow: EURO/Grey: FOR/Orange: GAC.

**Figure 4 materials-16-05604-f004:**
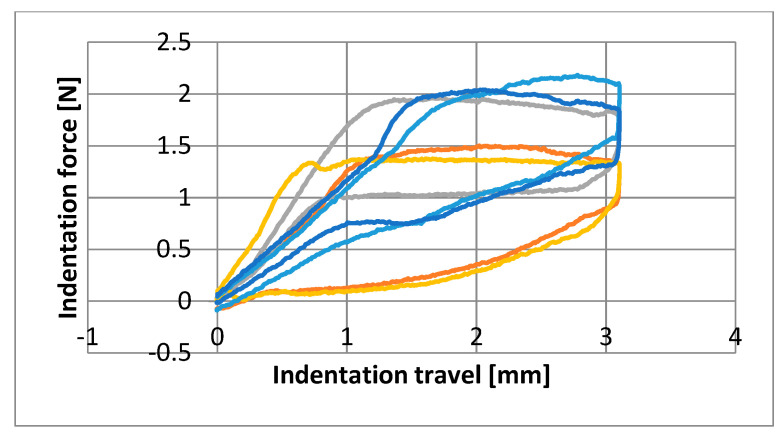
Diagram of the first specimen from each manufacturer obtained from archwires after three months of clinical use (T1). Dark blue: GC/Blue: 3M/Yellow: EURO/Grey: FOR/Orange: GAC.

**Figure 5 materials-16-05604-f005:**
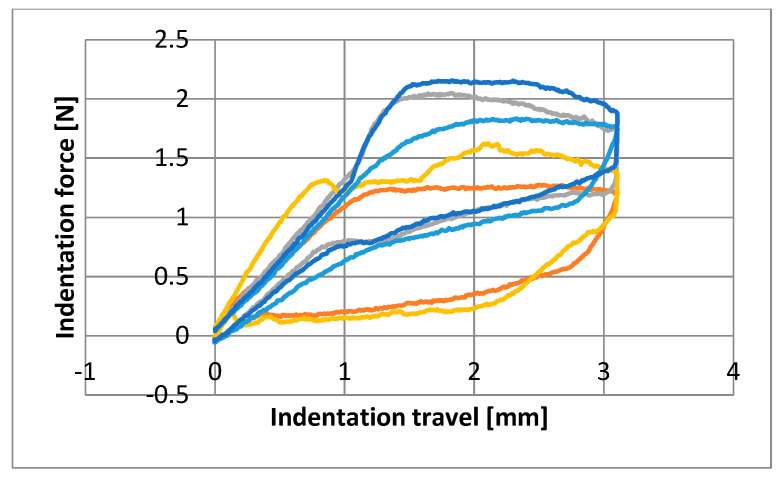
Diagram of the first specimen from each manufacturer obtained from archwires after sterilization (T2). Dark blue: GC/Blue: 3M/Yellow: EURO/Grey: FOR/Orange: GAC.

**Figure 6 materials-16-05604-f006:**
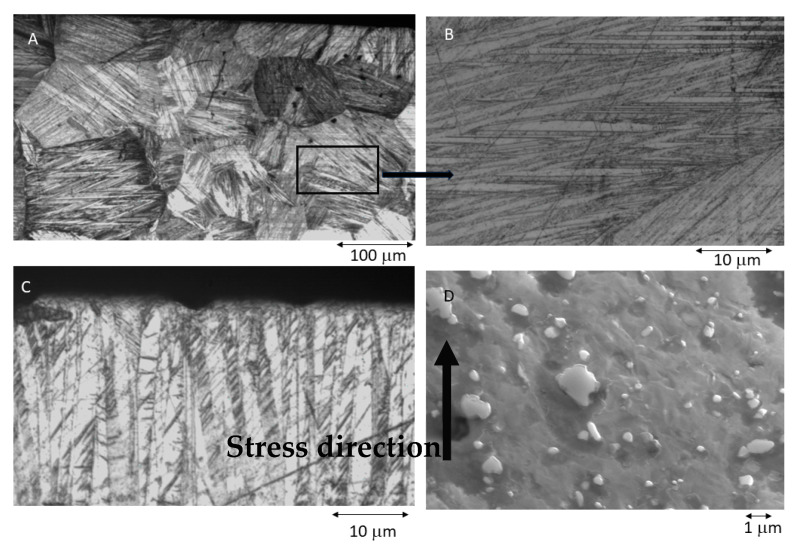
(**A**) Martensitic microstructure in equiatomic superelastic NiTi archwire. (**B**) Microstructure (**A**) at higher magnification. (**C**) Stabilized martensitic plates induced by stress. (**D**) Ti-rich precipitates on the austenitic matrix. In this case, the NiTi was treated at 550 °C for 60 min.

**Table 1 materials-16-05604-t001:** Archwire tested groups distributed by brands and batches.

Archwire Group	Archwire	Diameter	Manufacturer	Batch
GC-0 GC-1 GC-2	Nickel Titanium	0.016 inch	GC Orthodontics Europe GmbH, Germany	195415
3M-0 3M-1 3M-2	Nitinol^®^ SuperElastic	0.016 inch	Unitek, Monrovia, CA, USA	IE6ZG
EURO-0 EURO-1 EURO-2	Euro Ni-Ti Opto TH Plus	0.016 inch	ODS, Kisdorf, Germany	2001898 2001930
FOR-0 FOR-1 FOR-2	Titanol^®^ Superelastic	0.016 inch	Forestadent, Pforzheim, Germany	48055331 46055329
GAC-0 GAC-1 GAC-2	Sentalloy^®^ superelastic	0.016 inch	GAC, Grenoble, France	I760004 C760001

**Table 2 materials-16-05604-t002:** Mean values of Plateau 0.5 mm, Plateau 1 mm, and Plateau 2 mm measured for T0, T1, and T2 of each manufacturer. ANOVA Test. *p*-value 95% and clustering analysis of each group comparing T0–T1, T1–T2, and T0–T2. (NS Non-significant; ** *p* < 0.01, *** *p* < 0.001).

Groups	T0 ± SD	Cluster	T1 ± SD	Cluster	T2 ± SD	Cluster	T0–T1	Cluster	T1–T2	Cluster	T0–T2	Cluster
**Plateau 0.5 mm (N)**												
GAC	−1.92 (0.07)	a	−0.50 (0.17)	a	−0.56 (0.19)	a	−0.15 (0.14) **	c	−0.06 (0.08) ***	b	−0.21 (0.16) **	c
3M	−2.23 (0.13)	c	−1.56 (0.11)	b	−1.66 (0.11)	b	0.36 (0.13) ***	a	−0.10 (0.17) ***	b	0.26 (0.14) ***	b
ODS	−0.35 (0.18)	a	−0.47 (0.07)	a	−0.59 (0.07)	a	−0.18 (0.09) ***	c, d	−0.12 (0.08) ***	b	−0.30 (0.12) ***	c,d
GC	−1.68 (0.08)	b	−1.97 (0.14)	c	−2.07 (0.1)	c	−0.29 (0.17) ***	d	−0.10 (0.17) ***	b	−0.39 (0.11) ***	d
FOR	−0.29 (0.08)	d	−2.22 (0.14)	d	−0.59 (0.07)	a	0.01 (0.16) NS	b	1.63 (0.16) ***	a	1.64 (0.15) ***	a
**Plateau 1 mm (N)**												
GAC	−0.17 (0.09)	a	−0.36 (0.1)	a	−0.32 (0.10)	a	−0.19 (0.09) ***	d	0.04 (0.06) ***	b	−0.15 (0.09) ***	c
3M	−0.97 (0.02)	e	−0.81 (0.06)	b	−0.87 (0.06)	c	0.17 (0.05) ***	a	−0.07 (0.09) ***	c	0.10 (0.07) ***	b
ODS	−0.24 (0.06)	b	−0.34 (0.04)	a	−0.41 (0.04)	b	−0.1 (0.06) ***	c	−0.07 (0.06) ***	c	−0.17 (0.08) ***	c
GC	−0.57 (0.04)	c	−0.9 (0.07)	c	−0.93 (0.03)	d	−0.33 (0.07) ***	e	−0.04 (0.07) ***	c	−0.37 (0.05) ***	d
FOR	−0.86 (0.05)	d	−0.88 (0.07)	c	−0.41 (0.04)	b	−0.02 (0.06) NS	b	0.47 (0.08) ***	a	0.45 (0.07) ***	a
**Plateau 2 mm (N)**												
GAC	0.13 (0.02)	a	0.16 (0.026)	a	−0.06 (0.09)	a	0.03 (0.02) **	b	−0.22 (0.08) **	d	−0.20 (0.08) **	c
3M	−0.30 (0.03)	e	−0.13 (0.16)	b	−0.31 (0.09)	c	0.17 (0.16) ***	a	−0.18 (0.14) ***	c,d	−0.01 (0.10) NS	a,b
ODS	−0.18 (0.05)	c	−0.13 (0.04)	b	−0.24 (0.04)	b	0.05 (0.06) **	b	−0.11 (0.06) ***	c	−0.07 (0.07) ***	b
GC	−0.02 (0.04)	b	−0.28 (0.11)	c	−0.29 (0.06)	b,c	−0.27 (0.11) ***	d	−0.01 (0.11) ***	b	−0.28 (0.08) ***	d
FOR	−0.26 (0.04)	d	−0.36 (0.05)	c	−0.24 (0.04)	c	−0.10 (0.06) ***	c	0.12 (0.06) ***	a	0.02 (0.05) NS	a

**Table 3 materials-16-05604-t003:** Mean values of the results of the Fdef N (force delivered in Newtons) at 0.5 mm, 1 mm, and 2 mm measured for T0, T1, and T2 for each brand. ANOVA Test. *p*-value 95% and clustering analysis of each group at T0–T1, T1–T2, and T0–T2. (NS Non-significant; * *p* < 0.05; ** *p* < 0.01, *** *p* < 0.001).

GROUPS	T0 ± SD	Cluster	T1 ± SD	Cluster	T2 ± SD	Cluster	T0–T1	Cluster	T1–T2	Cluster	T0–T2	Cluster
**Force deflection 0.5 mm (N)**												
GAC	0.15 (0.10)	c,d	0.11 (0.09)	d	0.21 (0.08)	b	−0.04 (0.08) NS	a	0.10 (0.04) ***	a	0.15 (0.10) ***	c,d
3M	0.19 (0.03)	c	0.20 (0.05)	c	0.21 (0.04)	b	0.02 (0.07) NS	a	0.01 (0.05) ***	c	0.19 (0.03) NS	c
ODS	0.10 (0.04)	d	0.08 (0.04)	d	0.17 (0.03)	b	−0.01 (0.05) NS	a	0.09 (0.04) ***	a,b	0.10 (0.04) ***	d
GC	0.91 (0.03)	a	0.39 (0.04)	b	0.45 (0.05)	a	−0.52 (0.05) ***	b	0.06 (0.03) ***	b	0.91 (0.03) ***	a
FOR	0.52 (0.08)	b	0.50 (0.06)	a	0.17 (0.03)	b	−0.02 (0.09) NS	a	−0.33 (0.08) ***	d	0.52 (0.08) ***	b
**Force deflection 1 mm (N)**												
GAC	0.19 (0.09)	d	0.15 (0.08)	d	0.24 (0.08)	c	−0.04 (0.07) NS	b	0.10 (0.04) ***	a	0.19 (0.09) **	d
3M	0.57 (0.04)	c	0.51 (0.07)	c	0.52 (0.07)	b	−0.06 (0.08) *	b	0.01 (0.07) ***	b	0.57 (0.04) *	c
ODS	0.07 (0.03)	e	0.12 (0.05)	b	0.18 (0.03)	d	0.05 (0.05) **	a	0.06 (0.04) ***	a	0.07 (0.03) ***	e
GC	1.03 (0.04)	a	0.75 (0.06)	d	0.80 (0.05)	a	−0.29 (0.09) ***	c	0.05 (0.07) ***	a,b	1.03 (0.04) ***	a
FOR	0.95 (0.03)	b	0.93 (0.05)	a	0.18 (0.03)	d	−0.03 (0.07) *	b	−0.75 (0.07) ***	c	0.95 (0.03) ***	b
**Force deflection 2 mm (N)**												
GAC	0.38 (0.06)	c	0.35 (0.06)	b	0.40 (0.06)	c	−0.02 (0.05) NS	b,c	0.05 (0.04) ***	a	0.38 (0.06) NS	c
3M	0.97 (0.02)	b	0.97 (0.16)	a	0.84 (0.09)	b	0.01 (0.17) NS	b	−0.13 (0.13) ***	c	0.97 (0.02) ***	b
ODS	0.10 (0.03)	d	0.28 (0.04)	b	0.26 (0.02)	d	0.18 (0.05) ***	a	−0.02 (0.04) ***	b	0.10 (0.03) ***	d
GC	1.09 (0.04)	a	1.00 (0.07)	a	1.04 (0.05)	a	−0.09 (0.09) ***	c	0.04 (0.09) ***	a,b	1.09 (0.04) NS	a
FOR	1.09 (0.03)	a	0.99 (0.06)	a	0.26 (0.02)	d	−0.09 (0.07) ***	c	−0.73 (0.07) ***	d	1.09 (0.03) ***	a

## Data Availability

The authors can provide details of the research requirements by letter and comments under needs.

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
