# Peer review of "Properties of Superelastic Nickel–Titanium Wires after Clinical Use"

_materials, 2023, doi:10.3390/ma16165604_

Round 1
Reviewer 1 Report
Article “Properties of superelastic nickel-titanium wires after clinical use” by Ines Alcaraz and colleagues.
The article is devoted to the study of the titanium nickelide wires behavior under the bending loads. Nitinol wires are actively used both in medicine and in industry, and often they are subject to high demands on the accuracy of deformations under the various factors (for example, temperature or stresses). Therefore, an accurate understanding of the nitinol behavior under the given mechanical loads is important. Such studies should not be considered routine, because extensive statistics allow to analyze accurately the behavior of wires and predict correctly their behavior in practice. In this regard, the study presented in the paper is relevant.
The article as a whole has a correct structure and contains useful practical results. At the same time, there are a number of comments to the article that require its revision.
1. It is not clear from the introduction why it is necessary to carry out static bending tests. It is not clear why and how the obtained results will be used in practice in orthodontic treatment of teeth. It is difficult for people who are not familiar with the peculiarities of dental treatment to understand exactly where and how these nitinol wires are used in treatment practice. Perhaps some explanation should be added after the paragraph on lines 35-38.
2. Line 49 refers to Figure 2. This is incorrect. Firstly, up to this point in the text there was no reference to Figure 1. Secondly, the figure should be given immediately after the reference to it appears in the text. Thirdly, it is not clear which plateau in Figure 2 is referred to in the text. Further in the article, of course, everything becomes clear. But at the time of reading lines 49-51, nothing is clear. It is proposed either to transfer Figure 2 to the introduction with a more detailed description of all the parameters, or to make an additional scheme in the introduction.
3. A huge number of errors and typos are in the text. I will only point out some lines in which there are errors: 52, 66, 217, 220, 222, 238, 239. You need to read the article carefully again and correct all the typos.
4. Figure 1. I believe that depicting formulas in the form of a figure is not the best choice.
5. What is the reason for the presence of a vertical section (3.1 mm) on the deformation curve? Is it related to stress relaxation?
6. The main remark is on figures 3-5 and tables 2 and 3. Reading tables 2 and 3 is difficult due to the amount of information, but parsing figures 3-5 is even more difficult. It seems that it is not necessary to give all curves for all wire manufacturers. It is necessary to leave a number of graphs that reflect the meaning of the study, namely, on one graph, show the degradation of properties (T0 / T1 / T2) for one or two materials, on the second - the scattering for one / two materials, etc. Perhaps you should make separate graphs illustrating interesting patterns. Full data for all materials can be given in the appendix to the article. So far, in such a presentation of data (Figures 3-5), their meaning is lost. It is even difficult to compare the same material in states T0, T1 and T2, and this is the main idea of the article.
7. Nitinol is a very sensitive material to the variation of chemical composition. A change in the chemical composition by a few tenths of a percent can greatly change its properties. There is an opinion that if you take wires from the same manufacturer, but from different batches, the result may be different.
8. Figure 6. It is not clear, does Figure 6D show a wire with a different equiatomic composition compared to the wires studied in the paper? It is also not clear from the figures, where is the direction of mechanical stresses? The photos shown in Figure 6 do not look like a complete study of the structure.
___
I believe that the article needs to be corrected in accordance with these comments.
Reviewer 3 Report
I have read the research and is very interesting and novel. It is well-structured and presents interesting results, which are relevant to today's orthodontic treatments. Enlisted please find my suggestions regarding the article.

Round 2
Reviewer 1 Report
Article “Properties of superelastic nickel-titanium wires after clinical use” by Ines Alcaraz, et. al. (revised version review)
The authors corrected most of the comments that were given to the original version of the paper. The article became more reader-friendly, the authors were able to better present the results of their research.
I have to provide the following comments to the revised version of the article:
1. In the scheme shown in fig. 2, it is not clear what the inscriptions "1a" and "2a" mean, it is also necessary to indicate the point of force application.
2. If we consider the article from the side of the reader, then tables 2 and 3 are still too complex for analysis - average values, deviations, clusters, brands, different states of the alloy - all this information indicated within two tables. I suggest the authors to think again how to facilitate the perception of these tables.
I believe that after corrections the article can be published in "Materials".
Reviewer 2 Report
The authors have carefully revised and improved the article as per the suggestions of the reviewers; hence no further correction or suggestion is recommended.